Dark kinase annotation, mining, and visualization using the Protein Kinase Ontology

Soleymani Saber 1
Gravel Nathan 2
Huang Liang-Chin 2
Yeung Wayland 2
Bozorgi Elika 1
Bendzunas Nathaniel G. 3
Kochut Krzysztof J. 1 kkochut@uga.edu
Kannan Natarajan 2 3 nkannan@uga.edu
1 Department of Computer Science, University of Georgia , Athens, GA , United States
2 Institute of Bioinformatics, University of Georgia , Athens, GA , United States
3 Department of Biochemistry and Molecular Biology, University of Georgia , Athens, GA , United States
Góes-Neto Aristóteles
Electronic publication date: 2023 Dec 5
Publication date: 2023
Volume: 11
Electronic Location ID: e16087
Received 2023 Apr 5; Accepted 2023 Aug 22
Copyright: © 2023 Soleymani et al.
Copyright year: 2023
Copyright holder: Soleymani et al.
License: This is an open access article distributed under the terms of the Creative Commons Attribution License, which permits unrestricted use, distribution, reproduction and adaptation in any medium and for any purpose provided that it is properly attributed. For attribution, the original author(s), title, publication source (PeerJ) and either DOI or URL of the article must be cited.
License URL: https://creativecommons.org/licenses/by/4.0/

Keywords: Curation, User interface, Drug discovery, Cell signaling, Protein structure, Protein function, Network biology, Cancer genomes

Funding: National Institutes of Health U01CA239106 and R35 GM139656 This work was supported by the National Institutes of Health (U01CA239106 and R35 GM139656) to Natarajan Kannan. The funders had no role in study design, data collection and analysis, decision to publish, or preparation of the manuscript.

==============================
The Protein Kinase Ontology (ProKinO) is an integrated knowledge graph that conceptualizes the complex relationships among protein kinase sequence, structure, function, and disease in a human and machine-readable format. In this study, we have significantly expanded ProKinO by incorporating additional data on expression patterns and drug interactions. Furthermore, we have developed a completely new browser from the ground up to render the knowledge graph visible and interactive on the web. We have enriched ProKinO with new classes and relationships that capture information on kinase ligand binding sites, expression patterns, and functional features. These additions extend ProKinO’s capabilities as a discovery tool, enabling it to uncover novel insights about understudied members of the protein kinase family. We next demonstrate the application of ProKinO. Specifically, through graph mining and aggregate SPARQL queries, we identify the p21-activated protein kinase 5 (PAK5) as one of the most frequently mutated dark kinases in human cancers with abnormal expression in multiple cancers, including a previously unappreciated role in acute myeloid leukemia. We have identified recurrent oncogenic mutations in the PAK5 activation loop predicted to alter substrate binding and phosphorylation. Additionally, we have identified common ligand/drug binding residues in PAK family kinases, underscoring ProKinO’s potential application in drug discovery. The updated ontology browser and the addition of a web component, ProtVista, which enables interactive mining of kinase sequence annotations in 3D structures and Alphafold models, provide a valuable resource for the signaling community. The updated ProKinO database is accessible at https://prokino.uga.edu.

Introduction

The protein kinase gene family with nearly 535 human members (collectively called the human kinome) is a biomedically important gene family associated with many human diseases such as cancer, diabetes, Alzheimer’s disease, Parkinson’s disease, and inflammatory disorders. They make up one-third of all drug-related protein target discoveries in the pharmaceutical industry, with over 50 FDA-approved drugs developed since 2001 (Ferguson & Gray, 2018; Zhang, Yang & Gray, 2009). However, despite decades of research on the protein kinase family, our current knowledge of the kinome is skewed towards a subset of well-studied kinases with nearly one-third of the kinome largely understudied. These understudied kinases, collectively referred to as the “dark” kinome by the Knowledge Management Center (KMC) (Nguyen et al., 2017) within the Illuminating the Druggable Genome (IDG) consortium, constitute both active kinases and inactive pseudokinases, which lack one or more of the active site residues, but perform important scaffolding and regulatory roles in signaling pathways (Byrne, Foulkes & Eyers, 2017; Eyers, Keeshan & Kannan, 2017; Eyers & Murphy, 2013; Murphy, Mace & Eyers, 2017) and are druggable (Foulkes et al., 2018). Incomplete knowledge of the structure, function, and regulation of these understudied kinases and pseudokinases presents a major bottleneck for drug discovery efforts. While multiple initiatives are beginning to generate essential tools and resources to characterize dark kinases, integrative mining of these datasets is necessary to develop new testable hypotheses on dark kinase functions. However, integrative mining of protein kinase data is a challenge because of the diverse and disparate nature of protein kinase data sources and formats. Information on the structural and functional aspects of dark kinases, for example, is scattered in the literature posing unique challenges for researchers interested in formulating routine queries such as “disease mutations mapping to conserved structural and functional regions of the kinome” or “post-translational modifications (PTMs) in the activation loop of dark kinases.” Formulating such aggregate queries requires researchers to go through the often time-consuming and error-prone process of collating information from various data sources through customized computer programs, which results in duplication of efforts across laboratories, and does not scale well with the growing complexity and diversity of protein kinase data. For these reasons, the IDG consortium has developed a unified resource, Pharos, for collating diverse forms of information on druggable proteins, including protein kinases (Nguyen et al., 2017; Sheils et al., 2020, 2021). A focused Dark Kinase Knowledgebase has also been developed to make experimental data available on dark kinases to the broader research community (Berginski et al., 2021; Moret et al., 2021). While these unified resources provide a wide range of valuable information on druggable proteins, they offer limited data analytics capabilities in mining sequence and structural data. They do not conceptualize protein kinases’ detailed structural and functional knowledge in a practical and understandable way for protein kinase researchers. Thus, to accelerate the biochemical characterization of understudied dark kinases, a semantically meaningful and mineable representation of the kinase knowledge base is needed (Fig. 1).

Figure 1 The ProKinO architecture and work-flow.

(A) A subset of curated data sources used in ontology population. (B) A schematic of the ontology schema with classes (boxes) and relationships (lines) connecting the classes. (C) Applications for ontology browsing and navigation.

To semantically represent protein kinase data in ways protein kinase researchers use and understand, we previously reported the development of a focused protein kinase ontology, ProKinO (Gosal, Kannan & Kochut, 2011; Gosal, Kochut & Kannan, 2011; McSkimming et al., 2015), which integrates and conceptualizes diverse forms of protein kinase data in computer- and human-readable format (Fig. 2). The ontology is instantiated with curated data from internal and external sources and enables aggregate queries linking diverse forms of data in one place. ProKinO enables the generation of new knowledge regarding kinases and pathways altered in various cancer types, and new testable hypotheses regarding the structural and functional impact of disease mutations (Bailey et al., 2015; Cicenas & Cicenas, 2016; Goldberg et al., 2013; Gosal, Kannan & Kochut, 2011; Hu et al., 2015; Liu, Liang & Zhang, 2016; McClendon et al., 2014; McSkimming et al., 2016, 2014, 2015; Meharena et al., 2013; Mohanty et al., 2016; Nguyen et al., 2015; Oruganty & Kannan, 2013; Ruan & Kannan, 2015; Simonetti et al., 2014; Taylor et al., 2015; ManChon et al., 2014; Vazquez et al., 2016). For example, through iterative ProKinO queries and follow-up experimental studies, we identified oncogenic mutations associated with abnormal protein kinase activation and drug sensitivity (Lubner et al., 2017; McSkimming et al., 2016, 2015; Mohanty et al., 2016; Patani et al., 2016; Ruan & Kannan, 2015; Ruan, Katiyar & Kannan, 2017). We have also employed federated queries linking ProKinO with other widely used ontologies and resources such as the Protein Ontology (PRO), neXtProt, Reactome, and the Mouse Genome Informatics (MGI) to prioritize understudied dark kinases for functional studies and generate testable hypotheses regarding post-translational modification and cancer mutations (Huang et al., 2018).

Figure 2 Subset of the updated ProKinO schema with new classes and relationships.

The full schema can be accessed at http://prokino.uga.edu/. New classes are colored in cyan and pre-existing classes are colored in pink. Black arrows indicate new relationships introduced to connect the new classes.

While our preliminary studies have demonstrated the utility of ProKinO in hypothesis generation and knowledge discovery, to fully realize the impact of ProKinO in drug discovery and dark kinome mining, the ontology and the associated analytics tools need to be further developed to expand its scope and usability. For example, mutations at specific functional regions of the protein kinase domain, such as the gatekeeper and activation segments, are known to impact drug binding efficacies (Gajiwala et al., 2009; Yun et al., 2008). Likewise, kinase mRNA expression profiles strongly correlate with drug response (Benhar, Engelberg & Levitzki, 2002; Duncan et al., 2012; Kim, Song & Haura, 2009; Niepel et al., 2017). Thus, integrative mining of disease mutations with drug sensitivity profiles and expression patterns can provide new hypotheses/data for the development and administration of combinatorial drugs where multiple mutated kinases in distinct pathways can be targeted for drug repurposing (Erika et al., 2016; Li & Jones, 2012), as demonstrated by the repurposing of Gleevec for targeting c-kit kinase in Gastrointestinal tumors (Joensuu et al., 2001). Furthermore, the recent generation of structural models of various dark kinases using AlphaFold (Jumper et al., 2021) provides a new framework for generating new hypotheses by interactive mining and visualization of sequence annotations in the context of 3D models. However, the lack of interactive visualization tools to overlay sequence and functional annotations in 3D structural models presents a bottleneck in the effective use of AlphaFold models for function prediction. To address this and other challenges described above related to dark kinase mining and annotation, we have expanded ProKinO by including kinase expression data, as well as a variety of data related to ligand-motif interaction, and ligand response prediction (Huang et al., 2021). We have also significantly revamped the ProKinO browser through incorporation of new visualization tools for the interactive mining of sequence annotations in the context of experimentally determined 3D structures and AlphaFold models. We demonstrate the application of these new tools in dark kinase annotation and mining using the understudied p21-activated protein kinase 5, as an example. The updated ontology and browser provide a valuable resource for mining, visualizing, and annotating the dark kinome and pseudokinome.

Materials and Methods

Data sources

The ProKinO ontology includes data obtained from curated internal sources as well as external sources. Information from internal sources include annotations of kinase sequence and structural motifs retrieved from curated multiple sequence alignments. External sources are used for information related to kinase sequence and classification (KinBase & UniProt) (Bairoch et al., 2005; Manning et al., 2002) cancer mutations (COSMIC) (Tate et al., 2018), pathways (Reactome) (Croft et al., 2011) and three dimernsioanl structure (PDB) (Berman et al., 2000). The ontology is populated and updated on a regular basis using protocols described in previous studies (Gosal, Kochut & Kannan, 2011; McSkimming et al., 2016). Here, we describe further enhancements and additions to ProKinO through integration of data on kinase expression patterns and drug interactions, as described below. In a seperate significant project, we have identified and classified nearly 30,000 pseudokinases spanning over 1,300 organisms (Kwon et al., 2019). The schematic representation of the classification of kinases into groups, families, and subfamilies was already in place (Hanks & Hunter, 1995; Manning et al., 2002). Consequently, the addition of the pseudokinases and their classification was relatively simple. However, it significantly enhanced ProKinO as a comprehensive knowledge graph representing kinase-related data. The definition and nomenclature of several kinome-wide conserved motifs were standardized based on several previously published studies which describe the kinase structural features such as subdomains (Hanks & Hunter, 1995), regulatory spine/shell (Meharena et al., 2013), and catalytic spine (Hu et al., 2015). A subset of redundant or family-specific motifs were removed in the updated ontology and motif information on some of the atypical kinases such as ALPHAK2 is not included as they cannot be reliably aligned with canonical protein kinases. Portions of the text were previously published as a part of a preprint (Soleymani et al., 2022).

Ligand interactions

Information on kinase-ligand interactions were retrieved from the Kinase-Ligand Interaction Fingerprints and Structures (KLIFS) database (Kanev et al., 2021). The KLIFS database stores detailed drug-protein kinase interaction information derived from diverse (>2,900) structures of catalytic domains of human and mouse protein kinases deposited in the Protein Data Bank. In addition, KLIFS provides an application programming interface (API) for programmatic access to data related to chemicals and structural chemogenomics (Kanev et al., 2021). However, it lacks information regarding kinase pathways or diseases which prevents the user from investigating the effect of drug-mutant protein binding on downstream pathways or diseases. KLIFS annotations, which report PDB residue positions, were converted to UniProt residue numbering using PDBrenum (Faezov & Dunbrack, 2021), then converted to prototypic Protein Kinase A (PKA) numbering using Multiply Aligned Profiles for Global Alignment of Protein Sequences (MAPGAPS) (Neuwald, 2009). Entries that could not be mapped or did not map to the kinase domain were filtered out.

Ligand responses

We have also incorporated information on kinase drug sensitivity profile in the updated ProKinO. In particular, we retrieved drug dose response data for kinase-relevant ligands/drugs from the Genomics of Drug Sensitivity in Cancer (GDSC) (Yang et al., 2013). Kinase-relevant ligands are defined based on our previous study (Huang et al., 2020), which collected 143 small-molecule protein kinase inhibitors from GDSC based on four drug-target databases: DrugBank (Wishart et al., 2018), Therapeutic Target Database (Li et al., 2018), Pharos (Nguyen et al., 2017), and the Library of Integrated Network-Based Cellular Signatures (LINCS) Data Portal (Koleti et al., 2018). GDSC provides the half-maximal inhibitory concentration values (IC50) of these 143 ligands in 988 cancer cell lines.

Ligand activities

Ligand activities were retrieved from Pharos, a flagship resource (Nguyen et al., 2017) of the National Institutes of Health (NIH) Illuminating the Druggable Genome (IDG) program that includes data on small molecules, including approved drug data and bioassay data. Based on the protein classification (Lin et al., 2017), the drug targets in Pharos include kinases, ion channels, G-protein coupled receptors (GPCRs), and others. In this phase of the project, we decided to include the data relevant to ligand binding in kinases. Pharos integrates drug-target relationships from several resources, such as ChEMBL (Bühlmann & Reymond, 2020) and DrugCentral (Avram et al., 2021).

Expression data

An important part of our recent additions was kinase expression data. Genomic expression data (protein, RNA), as well as transcription factors and epigenomic associations, are among many facets of the data included in Pharos. Furthermore, the GDSC repository contains gene expression data (Affymetrix Human Genome U219 Array), as well. Additionally, COSMIC’s Cell Lines Project includes a significant amount of gene expression data, including kinase expression.

Dark kinases

Dark kinases were labeled based on the information from Dark Kinase Knowledgebase (Berginski et al., 2021).

Protein kinase knowledge graph: schema and data organization

The ProKinO ontology consists of classes, sub-classes, class types, relationships, relationship types, and constraints of protein kinase and related data (Fig. 2). The hierarchy connects all classes to the root, which is ProKinOEntity. Moreover, the schema defines types and constraints for the relationships. With such explicit and constrained schema, composing queries is more intuitive than conventional relational databases. In particular, to enable integrative mining of dark kinase expression data in the context of kinase sequence and structural features, we have introduced three new classes in ProKinO, the Ligand class (including its name, source, and chemical structure) and the following three related classes: (1) LigandInteraction, placed between the Ligand and (already existing) Motif classes to capture kinase-ligand binding and selectivity at the motif and residue level; (2) LigandActivity, placed between the Ligand and (already existing) Protein classes to represent kinases targeted by ligands (and drugs); and (3) LigandResponse, located between the Ligand and (already existing) Sample classes and representing ligand (and drug) sensitivity in kinases. To capture kinase expression, we added the GeneExpression relationship linking the Protein and Sample classes. The outline of the recently added classes and their relationships in ProKinO is illustrated as a UML class diagram, shown in Fig. 2.

ProKinO population

The ProKinO knowledge graph is automatically populated from several external and local data sources at regular intervals, as originally described (Gosal, Kochut & Kannan, 2011), ProKinO schema and the associated knowledge graph population software are routinely updated to incorporate additional sources of data such as pseudokinase and “dark” kinase classification and incorporating information on ligand interactions, ligand responses, ligand activities, kinase expression and associated object and datatype properties. We have been using the Protégé ontology editor for the schema creation and its subsequent modifications. The organization of the schema after these modifications is available at https://prokino.uga.edu/about.

The population software has been coded in Java and uses the Jena Framework. The population process is performed in several steps to add instances, their properties, and a combination of reading the prepared data from CSV, RDF, XML, and other file formats and accessing many remote data sources using their provided API (for example, Reactome’s REST API). Entity interconnections across data retrieved from different data sources are accomplished using UniProt identifiers, kinase names, and other accession identifiers. We modified the population software to create instances and properties for the newly added classes and relationships.

More specifically, using the KLIFS API, we retrieved the relevant kinases, ligands, and residue-level interaction data. The data was retrieved and then processed by custom Perl scripts. ProKinO ontology schema was modified, and ligands were included as new data, while interaction data (motifs) were either reconciled with the motifs already present in ProKinO or added as new, if not already there.

Similarly, the ligand response data was retrieved from GDSC and then processed by custom Perl scripts to create suitable CSV files. Additional ligands were included as new data, while the response data and the relevant samples were either reconciled with the samples already present in ProKinO or added as new, if not already there.

In order to populate the data on ligand activities, we retrieved from Pharos kinase-relevant ligands, as well as their binding data on targeted kinases, for example, IC50 values. This data was retrieved and then processed by custom Perl scripts to produce the necessary CSV files. Additional ligands, not included in the KLIFS dataset, were included as new data. All kinases targeted by ligands were already present in ProKinO, so they were reused in this step.

Data on kinase expression was first retrieved from Pharos, COSMIC, and GDSC. As before, the relevant kinases were already present in the ProKinO knowledge graph. The expression data was stored as individuals in the Expression class. Some of the relevant data about samples were already present in ProKinO, as we already had a significant amount of sample data from COSMIC. Additional samples were included as new data.

We reviewed and updated all the motifs already present in ProKinO. Furthermore, we updated the motif naming in cases where there were differences with the standard motif names.

Finally, we assembled an up-to-date list of dark kinases (Berginski et al., 2021) and added a Boolean datatype property, isDarkKinase, to identify them among all other kinases in the ProKinO knowledge graph.

Results/discussion

The expanded ontology and its knowledge graph provide a wealth of data unifying the information available on both well-studied (light) kinases and understudied (dark) kinases that serve as a unified resource for mining the kinome. The current version of ProKinO (version 65), includes 842 classes, 31 objects and 67 data properties, and over seven million individuals (knowledge graph nodes). ProKinO contains information on 153 dark kinases. A total of 137 dark kinases have information on structural motifs, 148 have disease mutations mapped to the kinase domain, 45 dark kinases have pathway information, and 26 are associated with specific reactions, as defined in Reactome.

Users can navigate the ontology using the ontology browser by searching for a specific kinase of interest or by performing aggregate SPARQL queries linking multiple forms of data. Currently 35 pre-written queries linking different data types can be executed using the ProKinO browser (http://prokino.uga.edu/queries). A user can also download the ontology or browse data based on organisms, functional domains, diseases, or kinase domain evolutionary hierarchy. Below, we focus on the application of complex SPARQL queries and the ProtVista visualization tools for the illumination of understudied dark kinases.

Mutation and expression of understudied PAK5 in human cancers

One possible way to prioritize dark kinases for functional studies is to ask the question, “which dark kinases are most mutated in human diseases, such as cancers?”. Typically answering this question would require collating and post-processing data from multiple resources such as COSMIC, Pharos, and the Dark Kinase Knowledgebase. However, with the updated Protein Kinase Ontology, these questions can be quickly answered using SPARQL. Having the “isDarkKinase” property within the Protein class and the RDF triples connecting the “Mutation”, “Sample” and “Sequence” classes, one can formulate aggregate queries requesting all dark kinases mutated in cancer samples. To avoid biases introduced by the length of protein/gene sequences (longer proteins tend to have more mutations), the query can be modified to normalize mutation counts by sequence length. Executing this modified query (Query 27, available at http://prokino.uga.edu/queries) displays the rank-ordered list of dark kinases based on mutational density. The top ten dark kinases with the highest mutational density are shown in Fig. 3A. Notably, the p21 activated kinase 5 (PAK5) is at the top of the list with a mutational density of 1.917, followed by CRK7 (1.054), PKACG (1.011), PSKH2 (1.01) TSSK1 (1.008), CK1A2 (0.991), ERK4 (0.966), DCLK3 (0.912), PKCT (0.894) and PAK3 (0.859). Having identified PAK5 as the most frequently mutated dark kinase in cancers, one can further query the ontology to explore the role of this kinase in various cancers. With the addition of the new “GeneExpression” class in ProKinO and the RDF triples connecting gene expression to the “Sample” and “Protein” classes (GeneExpression:InSample: Sample; GeneExpression:hasProtein: Protein), one can formulate queries for PAK5 expression in different samples (Fig. 3B). Rank ordering the samples based on PAK5 expression (Query 33) reveals cancer types such as adenocarcinoma (Zscore: 4,701.5) and hepatocellular carcinoma (Zscore: 2,038.3) that have previously been associated with abnormal PAK5 expression (Fang et al., 2014; Han et al., 2018; Huo et al., 2019; Zhang et al., 2017). However, the role of PAK5 in other cancer types such as acute myeloid leukemia (Zscore: 136.5) is relatively underappreciated (Quan et al., 2020). The identification of new cancer sub-types with dark kinase expression and regulation further exemplifies the use of ProKinO in knowledge discovery.

Figure 3 SPARQL query results for Query 27 and 33.

(A) Output of Query 27 requesting top 10 dark kinases with most mutations in different cancer types. The mutation counts are normalized by sequence length. (B) Output of Query 33 listing samples with abnormal PAK5 expression. The query also lists histology, cancer subtypes, regulation, and Z-score. Only a subset of the query results is shown because of space constraints.

Mutational hotspots in the activation loop of PAK5

Because ProKinO encodes a wealth of information on the structural and regulatory properties of multiple kinases, it can be used to generate mechanistic predictions on cancer mutation impact. We demonstrate this for the PAK kinases by asking the question “where are PAK5 mutations located in the protein kinase domain?” Using the RDF triples connecting the “Mutation”, “Motif” and “Sequence” classes (“Mutation: LocatedIn: Motif”; “Mutation:InSequence: Sequence”), one can formulate a query (Query 28) listing mutations in different structural regions/motifs of the PAK5 kinase domain. Examination of the query results reveals that the C-terminal substrate binding lobe (C-lobe) is more frequently mutated (320 mutations) relative to the N-terminal ATP binding lobe (N-lobe: 173 mutations) (Fig. 4A). Within the C-lobe, nearly 78 mutations map to the activation loop, which is known to play a critical role in substrate recognition and activation in a diverse array of kinases (Huse & Kuriyan, 2002; Kornev & Taylor, 2015; Oruganty & Kannan, 2012). Despite the prevalence of activation loop mutations in PAK5, there is currently no information on how these mutations impact PAK5 kinase structure and function. Nonetheless, based on the evolutionary relationships captured in ProKinO (based on the alignment of human kinases to the prototypic protein kinase A), one can formulate queries mapping mutations to specific aligned positions in the shared protein kinase domain. A query listing (wild type) WT type and mutant type residues in the activation loop of PAK5 and the equivalent aligned residue positions in PKA (Query 29) provides additional context for these mutations. For example, two distinct mutations map to residue P602PAK5 in the activation loop of PAK5 that structurally corresponds to a phosphorylatable residue, T197PKA, in PKA (Yonemoto et al., 1993). Having this context provides a testable hypothesis that S602 mutations in PAK5 impact kinase phosphorylation and regulation. Likewise, WT residue P607PAK5 is mutated in four distinct cancer samples and this position is equivalent to PKA residue P202PKA, which configures the activation loop for substrate recognition (Knighton et al., 1991). Thus, mutation of this critical residue is expected to impact substrate binding and activation loop phosphorylation in PAK5. Additional insights into these mutations can also be obtained by visualizing these residues in the context of the PAK5 AlphaFold models using the ProtVista viewer described below.

Figure 4 SPARQL query results for Query 28 and 29.

(A) Output of Query 28 listing the number of unique cancer-linked mutations at various structural locations of PAK5 kinase. (B) Output of Query 29 listing unique point mutations in the activation loop of PAK5 kinase. The query also lists the equivalent PKA position, disease type, primary site of the tissue sample, equivalent residue for the PKA positioning of PKA, and subtype of the tissue sample. Entries containing only one mutation per position were filtered from the original query. Only a subset of the query results is shown.

Insights into PAK5 ligand binding sites

With the conceptualization of new information related to kinase ligands, their mode of action and interaction with specific motifs in the kinase domain, new aggregate queries linking mutated kinases to drug sensitivity profiles, mode of action, and ligand binding sites can be performed using the updated ProKinO. For example, queries such as “list proteins and drugs or ligands interacting with the protein’s gatekeeper residue (GK.45)” (Query 31) and “list ligands targeting the Epidermal Growth Factor Receptor (EGFR) kinase and their mode of action” (Query 34) can be rapidly performed using the updated ProKinO ontology. We demonstrate the application of these new additions in the context of PAK5 by asking the question “what are the drugs targeting PAK family (PAK1-6) kinases?” Query 30 answers this question using the RDF triples connecting the “Ligand”, “Motif” and “Protein” classes (list triples) (Fig. 5). Examination of the query results indicates multiple drugs targeting PAK family kinases, including STAUROSPORINE and N2-((1R-2S)-2-AMINOCYCLOHEXYL) that bind to structurally equivalent residues/motifs in the ligand binding pocket of PAK4 and PAK5, respectively. The ligand binding sites, and associated interactions can also be visualized using the ProtVista viewer described below. Additional queries linking dark kinases to drug sensitivities, structural motifs, and pathways are listed on the ProKinO website at https://prokino.uga.edu/queries.

Figure 5 SPARQL query results for Query 30.

Output of Query 30 listing ligands interactions with each PAK family member (PAK1-6). It also includes motif names and positions of full sequence and PKA positioning. The output of Query 30 was rearranged to highlight the homology of PAK4 and PAK5 motif/ligand interactions and the figure highlights only a subset of the query results. Run SPARQL query for full results.

Visualization tools for dark kinase annotation and mining

To provide structural context for cancer mutations and to enable interactive mining of dark kinase sequence annotations in the context of 3D structures and predicted models from AlphaFold (Jumper et al., 2021; Tunyasuvunakool et al., 2021), we developed and incorporated a modified version of the ProtVista viewer in ProKinO. The viewer can be deployed for any protein kinase of interest by navigating to the Structure tab in the protein summary page and selecting either a PDB structure or AlphaFold model of interest. A snapshot of the ProtVista viewer displaying the AlphaFold model of PAK5 kinase is shown in Fig. 6. The ProtVista viewer uses an enhanced version of the Mol* viewer and the PDB web component (Watkins et al., 2017) to provide two-way interactive navigation between the 3D structure (Fig. 6A, top panel) and annotation viewer (Fig. 6A, bottom panel).

Figure 6 ProtVista viewer.

(A) AlphaFold2 model of PAK5 kinase is shown in the structure viewer (top panel). Sequence viewer with annotations are shown in the bottom panel. (B and C) Zoomed in view of structural interactions associated with S602 and P607 in PAK5 activation loop.

The annotation viewer consists of multiple tracks populated dynamically based on data from ProKinO and external sources such as UniProt. In addition, prediction confidence scores for AlphaFold models are displayed in the annotation viewer along with additional annotations such as conserved sequence motifs, subdomains, and structural motifs involved in kinase regulation. The annotation viewer also shows other annotations from external sources such as ligand binding sites and predicted functional sites. Users can hover over the residues on the 3D structure viewer to view the equivalent information on the annotation viewer and vice versa. For example, selecting the “activation loop” in the annotation viewer highlights the corresponding structural region in the AlphaFold model of PAK5 (Fig. 6A). Likewise, the selection of residues in the activation loop (S602 and P607) in the structure viewer highlights the annotations associated with these and interacting residues in the sequence viewer. Such interactive mining is expected to accelerate the functional characterization of dark kinases and provide new insights into disease mutations. For example, visualizing the interactions associated with S602 in the activation loop of PAK5 (Fig. 6B) indicates a hydrogen bonding interaction with R567, which is part of the conserved HRD motif (sequence annotation). Because the HRD-Arg is known to play a role in kinase regulation by stabilizing activation loop conformation (Huse & Kuriyan, 2002), it provides additional context for predicting the impact of S602 altering mutations. Likewise, examining the structural and sequence context of P607 interacting residues provides new insights into how the alteration of this residue might impact substrate binding and kinase regulation. Together, these examples highlight the value added by the ProtVista viewer in the visualization and annotation of mutations in dark kinases.

Conclusions

This work presents an updated version of the Protein Kinase Otology (ProKinO) for mining and annotating dark kinases. ProKinO was developed following FAIR (Findable, Accessible, Interoperable, and Reusable) principles (Wilkinson et al., 2016) and serves as an integrated knowledge graph for relating and conceptualizing diverse forms of disparate data related to protein kinase sequence, structure, function, regulation, and disease (cancer). We present a new ontology browser for navigating these data and demonstrate the application of aggregate SPARQL queries in uncovering new testable hypotheses regarding understudied kinase members. We also provide several pre-written SPARQL queries that can rapidly retrieve information related to protein kinase mutations, pathways, expression, and ligand binding sites. However, writing new queries requires prior knowledge of the ontology schema and the SPARQL query language, which most bench biologists may not have. To alleviate this challenge, we are currently building a graphical SPARQL query interface, which will intuitively enable query formulation through the navigation of the knowledge graph schema. We are also exploring the application of ProKinO for machine learning-based knowledge discovery and hypotheses generation.

We acknowledge members of the Kannan lab for their valuable comments and suggestions. We additionally acknowledge the various contributions to the databases we utilized made through the efforts of the IDG consortium and numerous investigator-initiated efforts.

Additional Information and Declarations

Competing Interests

Author Contributions

Data Availability

Natarajan Kannan is an Academic Editor for PeerJ.

Saber Soleymani conceived and designed the experiments, performed the experiments, analyzed the data, prepared figures and/or tables, authored or reviewed drafts of the article, updated the browser and ontology, and approved the final draft.

Nathan Gravel conceived and designed the experiments, performed the experiments, analyzed the data, prepared figures and/or tables, authored or reviewed drafts of the article, and approved the final draft.

Liang-Chin Huang performed the experiments, analyzed the data, authored or reviewed drafts of the article, and approved the final draft.

Wayland Yeung analyzed the data, authored or reviewed drafts of the article, and approved the final draft.

Elika Bozorgi analyzed the data, authored or reviewed drafts of the article, updated the browser and ontology, and approved the final draft.

Nathaniel G. Bendzunas analyzed the data, authored or reviewed drafts of the article, and approved the final draft.

Krzysztof J. Kochut conceived and designed the experiments, analyzed the data, prepared figures and/or tables, authored or reviewed drafts of the article, updated the browser and ontology, and approved the final draft.

Natarajan Kannan conceived and designed the experiments, analyzed the data, prepared figures and/or tables, authored or reviewed drafts of the article, and approved the final draft.

The following information was supplied regarding data availability:

The protein kinase ontology (ProKinO)’s current OWL file and previous versions are available at: https://prokino.uga.edu/downloads.html.

Future versions of the ontology also will be placed at the same address.

The ontology browser is available at https://prokino.uga.edu/browser. Users can save the results of queries in diagrams or other formats such as CSV.

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
