# Peer review of "Dark kinase annotation, mining, and visualization using the Protein Kinase Ontology"

_PeerJ, doi:10.7717/peerj.16087_

## Round 0.1 · original submission · Major Revisions

· Academic Editor

Major Revisions

Dear authors:
We have received three revisions of your paper: two of them accepting with Minor Revision and one of them rejecting your paper. As the Associate Editor of your submitted paper, I was waiting for a third revision exactly because we had very contrasting revisions initially (Minor Revision and Reject). My decision is for Major Revision, and I really expect that you strongly show clearly to reviewer 2 why your paper deserves to be published.

Reviewer 1 ·

Basic reporting

The authors meet the basic reporting criteria by writing in clear, unambiguous and professional English throughout their article. In the introduction, they give a very good level of background for the reader to understand what developments have been made already, with papers referenced appropriately, and the authors have clearly described how they further build on these developments. All of this places the reader in good stead for understanding the findings described later in the article. Figures are referred to in the main text clearly, and figures themselves are explained well by figure captions. The ontology browser itself is freely available to the public online to try the queries mentioned by the author in the paper.

Experimental design

This is an intriguing and original article describing a new tool related to Biological sciences that is in line with scope of the journal.
The purpose of the research is well-defined in the introduction of the manuscript, the authors state and identify the knowledge gap this will fill and how other researchers can utilise this for their own research.
Methods are described with sufficient detail, or references are made appropriately if details are not explored in this paper.

Validity of the findings

The conclusions are well stated and link back to the original aims of the research. The conclusions are supported by the example of PAK5 given by the authors, please see below for additional comments on other kinases.

Additional comments

I am glad to see the authors have addressed potential usability issues for SPARQL queries in the conclusions, and that they are actively working on this. User-friendliness of these queries became a prominent question when reading the manuscript and would be a major hurdle for the accessibility of this tool for many users.


Questions:

Is integration of all the sources mentioned from lines 137-138 into ProKinO an ongoing process? The next questions could relate to this point.

For three of the ten dark kinases PSKH2, DCLK3, ALPHAK2 listed in lines 290-293, when trying to run query 28 on them mentioned in line 313 (to understand which kinase features mutations frequently occur in), no data comes up for these other dark kinases. Why might this be the case, since these proteins are listed as having mutations in the COSMIC database?

In lines 359-361 the authors note that structures can be viewed for any protein kinase in the ProKinO database by selecting either the PDB or AlphaFold structure. For many kinases however (e.g. for PKACA, VGFR1, ATR, FRAP, and several others), the structure dropdown list includes only an AlphaFold2 structure or the AF2 structure and one other structure, but several other experimentally-determined structures are noted in UniProt (which the authors mention was used as a source for ProKinO earlier in the paper). I understand that this is not directly related to dark kinases, but I wondered why these other experimentally-determined structures for light kinases are not available in the dropdown list?


Minor points:

It is exciting to see a tool developed where mutations and other features can be directly mapped onto the structure, and for this to be available within one resource for all kinases. There is just a minor point regarding the 3D structure viewer ProtVista. Due to the limited size of the browser window (at least on my laptop, though this would likely occur for other users too), when trying to match up features from the annotation viewer with the 3D structure (or vice versa), it is not always possible to view the structure at the same time as features that are lower down in annotations list. Would it be possible to have a feature to pin the 3D viewer window, or to keep the annotations viewer in a scrolling panel that is fixed below the 3D viewer? Please let me know if I have missed this feature and it is already available.


Line 329 “P604(PAK5)” shows a different font family used for P604 than the rest of the text. Additionally, do the authors mean P607 instead of P604?

In line 383 again the authors write P604, when they seem to mean P607 (from examining figure 4).


Other minor structure, grammar and punctuation points that were noticed during the review (if they have not been spotted already):

Line 47: “is one of the biomedically important gene families with direct associations with” could be reworded as “is a biomedically important gene family directly associated with” to prevent repetition of “with”.

Line 52: Should be “one-third” with a hyphen, to be consistent with its first use in line 49.

Line 106: A comma does not seem necessary after “the ontology”.

Line 324: The authors note “WT type”, and though it is a common abbreviation, it would be good to elaborate with the full name in brackets (wild type) in the first instance.

Line 351: Protvista should be ProtVisa with a capitalised V for consistency, the same applies to the caption title of Figure 6.

Line 391, the reference seems out of place, and would be better placed after the word “principles” rather than after “serves”.

Reviewer 2 ·

Basic reporting

The authors present an update of their work (Gosal et al. 2011a; Gosal et al. 2011b; McSkimming et al. 2015), the manuscript is well-organized and referenced.

Experimental design

NA

Validity of the findings

This is an update of a previous resource so it doesn't apply.

Additional comments

I don't think that this update is significant enough to justify a new manuscript given that authors have published several papers about this resource in the past.

·

Basic reporting

This paper describes an updated and expanded version of ProKinO, the Protein Kinase Ontology. Previously described in two publications -- (1) Gosar et al., 2011, and (2) McSkimming et al., 2015 -- this paper focuses on these enhancements in data content and analysis tools.

Experimental design

Not applicable. The paper describes a database and computational system.

Validity of the findings

In the context of a designed experiment, not applicable. In the context of retrospective knowledge discovery employing the described ProKinO system, the findings largely relate to the roles of PAK5 (p21 activated kinase 5) in cancers. Validity in this context may be defined and interpreted to refer to the logic and rigor of the knowledge inference methodology, and hence the reliability of the findings, given the input data and theoretical assumptions. Interpreted thus, the findings are well justified and valid.

Additional comments

This paper is well scoped, well justified, and well written. ProKinO is an established and valued resource for the biomedical research community, as indicated by the prior two ProKinO publications, and citations thereof. Thus my comments and suggestions are all minor.

1. The "Ligand responses" section refers to dose-response data, also known as "concentration-response" data. To my knowledge, these terms are conventional, but "dose and response data" is not, or much less so, hence less comprehensible to most readers.

2. "WT type" means "wild-type type"? Maybe "wild type (WT)" is an improvement.

3. Figure 1, the ProKinO architecture and work-flow, is organized into "Data Sources", "Ontology Population", and "Ontology Browser". Sparql is specified as the query language. This suggests that the data is stored as RDF, but this is not specified. I presume the ontology is represented via OWL and the instances as an RDF triple-store, but this is not specified. I suggest this be clarified. If not specified in this paper, with references to relevant online documentation.

---

## Round 0.2 · accepted · Accept

· Academic Editor

Accept

The authors have addressed all of the reviewers' comments and suggestions; therefore, I consider that the paper is ready to be accepted.

Reviewer 1 ·

Basic reporting

No comment

Experimental design

No comment

Validity of the findings

No comment

Additional comments

The authors have done well to address all of my initial questions and comments, and I suggest that the paper be accepted, there are only very minor changes to be made to the text (small typos/errors that would not need re-reviewing, that have been included in a separate PDF).

Annotated reviews are not available for download in order to protect the identity of reviewers who chose to remain anonymous.

Reviewer 2 ·

Basic reporting

The authors have address all my comments.

Experimental design

no comment

Validity of the findings

no comment

Additional comments

no comment

·

Basic reporting

This paper describes an updated and expanded version of ProKinO, the Protein Kinase Ontology. Previously described in two publications -- (1) Gosar et al., 2011, and (2) McSkimming et al., 2015 -- this paper focuses on these enhancements in data content and analysis tools.

Experimental design

Not applicable. The paper describes a database and computational system.

Validity of the findings

In the context of a designed experiment, not applicable. In the context of retrospective knowledge discovery employing the described ProKinO system, the findings largely relate to the roles of PAK5 (p21 activated kinase 5) in cancers. Validity in this context may be defined and interpreted to refer to the logic and rigor of the knowledge inference methodology, and hence the reliability of the findings, given the input data and theoretical assumptions. Interpreted thus, the findings are well justified and valid.

Additional comments

The concerns and comments of this reviewer pertaining to the initial manuscript have been addressed fully by the author responses and revisions to the manuscript.